# Transformers Can Perform Distributionally-robust Optimisation through In-context Learning

**Taeyoung Kim**[1]   **Hongseok Yang**[1]

## Abstract

Recent empirical and theoretical studies have shown that through in-context learning, transformers can solve various simple machine-learning problems such as linear regression and decision-forest prediction. We extend this line of research by analysing the power of transformers' in-context learning. We experimentally show that even in the presence of multiple types of perturbations, transformers can in-context learn a range of function classes. This means that transformers can perform the distributionally-robust optimisation (DRO) for those function classes when trained with appropriate in-context learning tasks. Our experiments include problems studied in the DRO community, which consider a single type of perturbations specified in terms of either total-variation distance or Wasserstein distance or the combination of multiple types of perturbations. Our experimental findings show that transformers can solve the DRO problems in all these cases. We also show that while standard algorithms for DRO are usually limited to linear models, through in-context learning, transformers can do DRO for non-linear models, such as kernel regression models and shallow neural networks.

## 1. Introduction

The in-context learning of transformers has been a topic of active research in recent years due to its surprising ability to improve the reasoning ability of language models (Brown et al., 2020; Hao et al., 2022; Wei et al., 2022). In particular, Garg et al. (2022) proposed a framework to study the inner working mechanism of transformers' in-context learning through the problem of learning a function class; given an unknown target function from the class and a training set

[1]School of Computing, KAIST, Daejeon, Korea. Correspondence to: Hongseok Yang <hongseok.yang@kaist.ac.kr>.

*Proceedings of the 1st Workshop on In-Context Learning at the 41st International Conference on Machine Learning*, Vienna, Austria. 2024. Copyright 2024 by the author(s).

generated by the target function, they encoded all of the input-output pairs in the training set and a query input as a sequence of tokens, gave it to a transformer, and decoded the next token predicted by the transformer as the output for the query. They found that transformers can be trained to in-context learn a range of simple function classes, such as classes of linear functions, shallow neural networks, and decision forests.

In this paper, we extend Garg et al. (2022)'s analysis to the setting of robust learning. Using their framework, we experimentally show that properly trained transformers can in-context learn a range of function classes even in the presence of perturbations, thereby performing a form of distributionally-robust optimization (DRO) during inference. Our analysis considers multiple types of perturbations and function classes, and some of them go beyond what has been traditionally considered by the DRO community.

Another goal of our work is to show the potential of transformers for solving new types of DRO problems through in-context learning and thus advancing the DRO research. DRO is commonly formulated as the min-max optimisation problem

$$\min_{\theta \in \Theta} \max_{P \in \mathcal{P}} \mathbb{E}_{z \sim P}[L(\theta; z)] \tag{1}$$

where $\mathcal{P}$ is the set of distributions close to the true distribution, $\theta$ is the parameter to be optimised, and $L$ is the loss function. While often coming with solid theoretical analysis, the existing works for DRO have some important drawbacks that block their practical applications. First, the existing techniques for DRO in machine learning are usually limited to linear models. Even the kernel regression goes beyond the scope of these techniques. Second, although interesting theoretical results about DRO exist, they are subtle and cannot be easily used in practice due to the failure of the assumptions of these results. The proofs of the results utilise advanced probability theory or optimisation theory heavily, so adapting them to new settings in practice is also difficult. Finally, the approaches for solving DRO problems with theoretical guarantees often have high time complexity, and cannot be applied for large-scale instances in practice, especially when the input dimension is high. For instance, Zhu et al. (2021a)'s algorithm on linear regression for total-variation (TV) distance DRO has the $O(N^2 d^4 + d^{12})$ time

complexity for the input dimension $d$ and the number of points $N$.

Meanwhile, transformers suffer from these drawbacks less severely. Even when a given DRO problem is over non-linear models, such as kernel-based models and neural networks, the training of a transformer for this problem is identical to that for linear models. So, at least in principle, we can train the transformer for the DRO problem over non-linear models, and apply the trained transformer to solve an instance of the problem through in-context learning. Furthermore, the issue of high time complexity does not arise for transformers, because the inference of transformers has quadratic time complexity in the length of the input sequence and the input dimension. Of course, the remaining question is whether trained transformers can actually solve DRO problems well in practice. Our paper shows some positive experimental results in this direction.

This paper is structured as follows. In Section 2, we describe the design of our experiments, including the perturbations considered and the generation of training sets for in-context learning. In Section 3, we report the findings of the experiments. In Section 4, we provide the preliminary interpretation for the success of transformers' in-context learning for DRO. In Section 5, we review the related works on in-context learning and DRO. Finally, in Section 6, we conclude our work and suggest possible future works.

## 2. Design of Experiments

In our experiments, we use the scheme diagramatically described in Figure 1 to generate a training dataset for DRO and train a transformer on the dataset. First, we sample target functions $f_1^*, \ldots, f_M^*$ independently from some distribution on a given function class. Next, for each $f_k^{(*)}$, we generate a training dataset $\{(\tilde{x}^{(k,i)}, \tilde{y}^{(k,i)})\}_{i=1}^N$ of perturbed input-output pairs as follows. We sample $N$ clean input-output pairs $\{(x^{(k,i)}, y^{(k,i)})\}_{i=1}^N$ where $x^{(k,i)}$ is sampled independently from some distribution $p$ and $y^{(k,i)}$ is set to the output $f_k^*(x^{(k,i)})$ with an additive noise $\epsilon^{(k,i)}$. We then perturb these sequences to get the training dataset $\{(\tilde{x}^{(k,i)}, \tilde{y}^{(k,i)})\}_{i=1}^N$. Third, we represent each of the generated datasets as a sequence, and feed these sequences to the transformer so as to obtain the sequences of predicted outputs $\{f_x^{(k,i)}\}_{(k,i)=(1,1)}^{(M,N)}$. Finally, for each $k$, the predicted outputs $\{f_x^{(k,i)}\}_{i=1}^N$ are compared with the clean outputs $\{f_k^*(\tilde{x}^{(k,i)})\}_{i=1}^N$ on the perturbed inputs $\{\tilde{x}^{(k,i)}\}_{i=1}^N$, through a loss function $\ell$. Note that these clean outputs can be computed since the $f_k^*$ are available during training. The sum of all the computed losses over all $M$ sampled target functions is used as the training loss for the transformer.

As one can see, there are several placeholders in the above scheme, in particular, the input distribution $p$, the class of

target functions, the distribution over this class, and the type of perturbations. In our experiments, the input distribution $p$ is set to be a multivariate normal distribution whose mean and covariance matrix are sampled randomly from the standard normal distribution and the Wishart distribution, respectively.

For the class of target functions, we consider the following three classes: linear functions, functions based on the RBF kernel (more precisely, functions in the reproducing kernel Hilbert space of the RBF kernel), and shallow neural networks. The choice of the distribution over the target functions depends on the function class. In the case of linear functions from $\mathbb{R}^d$ to $\mathbb{R}$, we sample a weight $w \in \mathbb{R}^d$ from a standard multivariate normal distribution, and define the target linear function by $f^*(x) = \langle w, x \rangle$. For the next case of kernel-based functions, we use the approximation of the RBF kernel with random Fourier features, and sample functions from the reproducing kernel Hilbert space (RKHS) of the RBF kernel approximately. Concretely, we approximate the RBF kernel $k : \mathbb{R}^d \times \mathbb{R}^d \to \mathbb{R}$ by

$$K(x, x') = \langle \Psi(x), \Psi(x') \rangle$$

where $\Psi : \mathbb{R}^d \to \mathbb{R}^F$ is the random feature map defined by

$$\Psi(x)_i = \sqrt{\frac{2}{F}} \cos(\langle z_i, x \rangle + b_i) \tag{2}$$

with independent samples $b_i \sim \text{Unif}([0, 2\pi])$ and $z_i \sim \mathcal{N}(0, I_d)$ for $i = 1, \ldots, F$. Then, we sample a target function $f^*$ by sampling $w \sim \mathcal{N}(0, \sigma_{\mathcal{H}}^2 I_F / F)$ for some appropriate choice of $\sigma_{\mathcal{H}}$, and setting $f^*(x) = \langle w, \Psi(x) \rangle$. Finally, for the shallow neural networks, we consider a shallow ReLU network $f^*$ with 100 neurons:

$$f^*(x) = \frac{1}{\sqrt{100}} \sum_{i=1}^{100} a_i \text{ReLU}(\langle w_i, x \rangle + b_i) \tag{3}$$

and sample such a network by randomly setting its parameters: $w_i \sim \mathcal{N}(0, \sigma^2 I_d / d)$, $a_i \sim \mathcal{N}(0, 2)$, and $b_i \sim \mathcal{N}(0, 1)$.

For the type of perturbations, we consider three options. Let $\mu$ be the distribution of the clean input-output pair for a fixed target function, and let $\nu$ be the distribution of the perturbed input-output pair for the same function. The first type is the perturbations specified in terms of the total-variation (TV) distance. It fixes the distribution set $\mathcal{P}$ in Equation 1 to

$$\mathcal{P} = \{\nu : \text{TV}(\mu, \nu) \leq \epsilon\}$$

for some $\epsilon > 0$, where $\text{TV}(\mu, \nu)$ is the total-variation distance between $\mu$ and $\nu$ and is defined as the superimum of $|\mu(A) - \nu(A)|$ over all measurable subsets $A \in \mathbb{R}^d \times \mathbb{R}$. Intuitively, this type means that the up-to-$\epsilon$ proportion of the clean input-output pairs in the training dataset can be perturbed arbitrarily. The second type is the perturbations

specified in terms of the Wasserstein-2 distance, which is defined by

$$W_2(\mu, \nu) = \inf_{\pi \in \mathcal{P}(\mu \times \nu)} \left( \int_{(\mathbb{R}^d \times \mathbb{R})^2} d(x, x')^2 d\pi(x, x') \right)^{1/2}$$

where $d(x, y)$ is the metric on $\mathbb{R}^d \times \mathbb{R}$ (i.e., the space of input-output pairs), and $\mathcal{P}(\mu \times \nu)$ is the set of particular distributions on $(\mathbb{R}^d \times \mathbb{R})^2$ called couplings between $\mu$ and $\nu$. Now the distribution set $\mathcal{P}$ in Equation 1 is set to

$$\mathcal{P} = \{\nu : W_2(\mu, \nu) \leq \rho\}$$

for some $\rho > 0$. Informally, the upper bound by $\rho$ in this case limits the amount that they are perturbed, because of the incorporation of the distance $d$ in the definition of $W_2$. The last type of perturbations we consider is the combination of the first two types, which has been studied in the outlier-robust Wasserstein DRO by Nietert et al. (2023). Specifically, we set

$$\mathcal{P} = \{\nu : W_2^\epsilon(\mu, \nu) \leq \rho\}$$

for some $\epsilon, \rho > 0$, where $W_2^\epsilon$ is the $\epsilon$-outlier-robust Wasserstein-2 distance defined by

$$W_2^\epsilon(\mu, \nu) = \inf_{\substack{\mu' \in \Pr(\mathbb{R}^d \times \mathbb{R}) \\ TV(\mu, \mu') \leq \epsilon}} W_2(\mu', \nu).$$

Informally, the DRO under this type of perturbation considers the scenario where the up-to-$\epsilon$ portion of the input-output pairs in the training dataset is changed arbitrary first and then the further perturbations but with $\rho$-limited changes (in the sense of Wasserstein-2 distance) occur in the resulting pairs.

In all of these three types of perturbations, generating a perturbed dataset from a clean dataset efficiently is not straightforward due to the upper bound in the definition of $\mathcal{P}$. In Appendix C, we explain how we address this issue in our experiments.

## 3. Experimental Results

We now report the findings of the experiments on DRO and the in-context learning of transformers, which were conducted according to the setup described in Section 2 and with GPT2 as the transformer architecture. For baselines, we consider several existing learning algorithms, including the ones that are not designed for DRO as well as the 3-nearest neighbour algorithm which is known to be robust to perturbations (Wang et al., 2018) for diverse classes of functions.

For the distribution set in Equation 1, we consider the following choices: when $\mu$ is an unperturbed distribution of

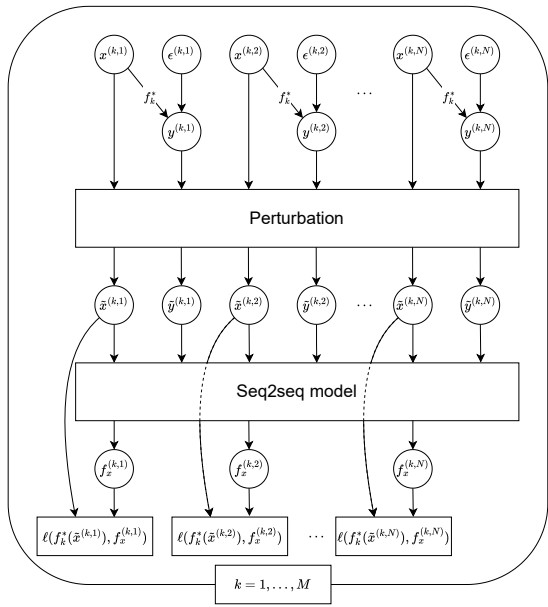

Figure 1. Visualisation of the scheme for generating a perturbed training dataset and training a transformer for this dataset for its in-context learning.

the input-output pairs,

$$\mathcal{P}_{\text{TV}} = \{\nu : TV(\mu, \nu) \leq 0.1\},$$
$$\mathcal{P}_{\text{W}} = \{\nu : W_2(\mu, \nu) \leq 1.0\},$$
$$\mathcal{P}_{\text{TVW}} = \{\nu : W_2^{0.05}(\mu, \nu) \leq 0.5\}.$$

These sets formalise the three types of perturbations based on bounded TV distance, bounded Wasserstein-2 distance, and bounded outlier-robust Wasserstein-2 distance. We use gpt2_tv, gpt2_w, gpt2_tvw to denote the transformers trained for these three types of perturbations. For the function classes, we chose both $\sigma_\mathcal{H}$ in the initialisation of Equation 2 and $\sigma$ in Equation 3 to be $\sqrt{20}$.

Figure 2 shows the results of the experiments on the problem of learning a linear function under the three types of perturbations and the different sizes of the in-context samples. All of gpt2_tv, gpt2_w, and gpt2_tvw are tested in all three cases of perturbations. The baselines are vanilla linear regression (LinearRegression), ridge regression (RidgeRegression), L1-regularised linear regression (LassoRegression), WassDRO proposed by Blanchet et al. (2019) but without cost learning, and the simple averaging estimator $w = \frac{1}{N} \sum_{i=1}^N x^{(i)} y^{(i)}$ (Averaging). Note that all the baselines except the Wass-DRO perform poorly, and also that in all the three types of perturbations, the trained transformer trained on the corresponding perturbation type outperforms all the baselines in

this DRO problem. Another interesting observation comes from the transformer gpt2_tvw trained on the third type of perturbations. This transformer performs well in all three types of perturbations, even though the strength of perturbations in the third type can be weaker than those perturbations in the first two types due to the use of tighter upper bounds 0.05 and 0.5 in $\mathcal{P}_{\text{TVW}}$ than 0.1 and 1.0 in $\mathcal{P}_{\text{TV}}$ and $\mathcal{P}_{\text{W}}$. This indicates the possible out-of-distribution generalisation of gpt2_tvw against the perturbations.

In Section B of the appendix, we provide additional experiments on this out-of-distribution generalisation, where we train our model on different levels of perturbation, and compare the performance of the model trained on different level of perturbation. We observe that the model trained on a moderate level of perturbation matches the performance of the model trained on a weaker or stronger level of perturbation, which indicates the out-of-distribution generalisation of the model.

We want to point out that the trained transformers not only outperform the baselines in terms of accuracy but also in terms of speed. In the evaluation of the transformers and the baselines, we use 12,800 instances of target functions to evaluate their performance. The inference of the trained transformers takes less than 1 minute, while the most promising baseline WassDRO takes around an hour. The reason for this superior performance is that these transformers can be parallelised on the GPUs, while the baselines can not, and also the inference of these transformers has the quadratic time complexity in the length of the input sequence, while most of the baselines have higher time complexities in the number of training samples and the dimension of the input space.

Figure 3 shows the results of the experiments on the class of functions based on the RBF kernel. The baselines in this case are the kernel regression with and without ridge regularisation (KernelRegression, KernelRidgeRegression), the kernel-based averaging estimator $f(x) = \frac{1}{N} \sum_{i=1}^{N} y^{(i)} K(x, x^{(i)})$ (KernelAveraging), and the nearest-neighbour algorithm (NearestNeighbours). Among the three kernel-based baselines, only the kernel regression with regularisation has the theoretically-guaranteed robustness to perturbations, but it performs worse than the nearest-neighbour algorithm. Again, in each of the three perturbation types, the transformer trained on the corresponding perturbation type outperforms all the baselines.

Figure 4 shows the experimental results on the class of shallow neural networks. In this case, we use only the nearest-neighbour algorithm as the baseline, because the other baselines are not designed for the DRO with neural networks and are expected to perform poorly. We do not try the neural network baseline where a separate neural network is trained for each of the 12,800 DRO instances.

This is because of the high computational cost that goes well beyond our computational resources. The results are consistent with those for the other two function classes; the trained transformers outperform the nearest-neighbour algorithm on the DRO under all three types of perturbations.

## 4. Transformers and DRO

As explained in Section 3, the transformer trained with the right type of perturbations outperforms all the baselines in our experiments, demonstrating that transformers can solve DRO problems through in-context learning. In this section, we provide a possible explanation for this success of transformers. Specifically we report the results of an additional experiment on a simple robust-learning problem, where we compare the in-context learning of transformers with an ideal baseline. The results show that the trained transformer and this ideal baseline performs nearly the same, indicating the possible similarity between the inner workings of the trained transformer and that of the ideal baseline.

Our additional experiment is on the problem of estimating the mean and variance of an unknown multivariate diagonal normal distribution from perturbed samples. We consider only perturbations specified in terms of the TV distance. This means that the perturbation works by arbitrarily changing the up-to-$\epsilon$ proportion of the samples in the clean training dataset. As a result, at least $(1 - \epsilon)N$ samples in the dataset are still clean. Except the choice of this new prediction problem, the rest of the experimental setting is identical to the one used in our previous experiments.

An ideal algorithm for solving this DRO problem is to identify the perturbed samples in a given dataset and and predict mean and variance only from the other clean samples. This algorithm cannot be used in practice due to the assumption that it knows which samples are perturbed and which are not. But it can be tried in our experiment where we know all the ground truths. Our baseline is this algorithm where the prediction of mean and variance is done by a transformer trained for solving this mean and variance prediction task through in-context learning but with clean datasets.

Figure 5 describes the results of our experiment. We include three naive baselines in this experiment, the empirical average of mean and variance (Averaging), and two robust estimators where we trim items far from the mean in terms of entries (EntryTrim) or the norm (NormTrim). We also include the ideal empirical average, which takes the average of the clean samples only. As expected, both gpt2 and gpt2_ideal outperform the baselines, and even the Averaging_ideal, which indicates that the gpt2 model also uses the prior distribution of the training dataset and uses this info to obtain the better prediction. If we compare the gpt2 and gpt2_ideal, it shows that the transformer trained for this

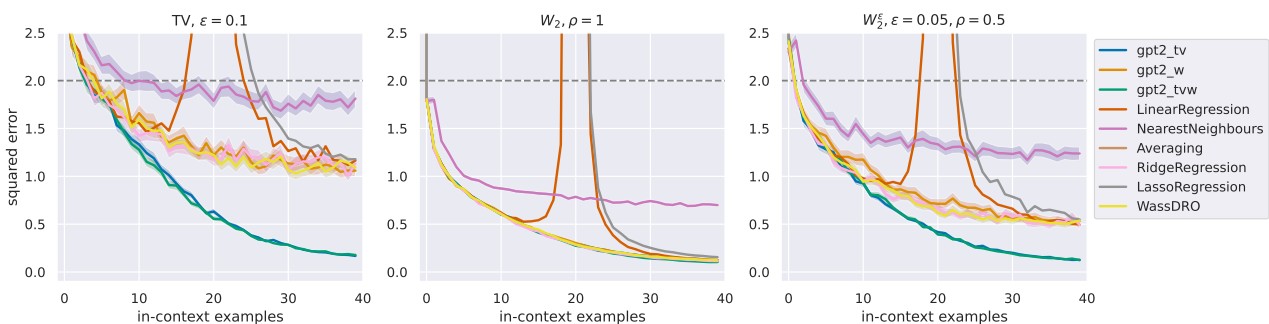

*Figure 2.* Results of the experiments on the class of linear functions. The x-axis is the number of in-context samples, and the y-axis is the L2 loss of the predictor. The dashed grey line is the loss of the trivial predictor, $f = 0$. (Left) Result of the TV distance-based DRO. (Middle) Result of the Wasserstein distance-based DRO. (Right) Result of the outlier-robust Wasserstein distance-based DRO.

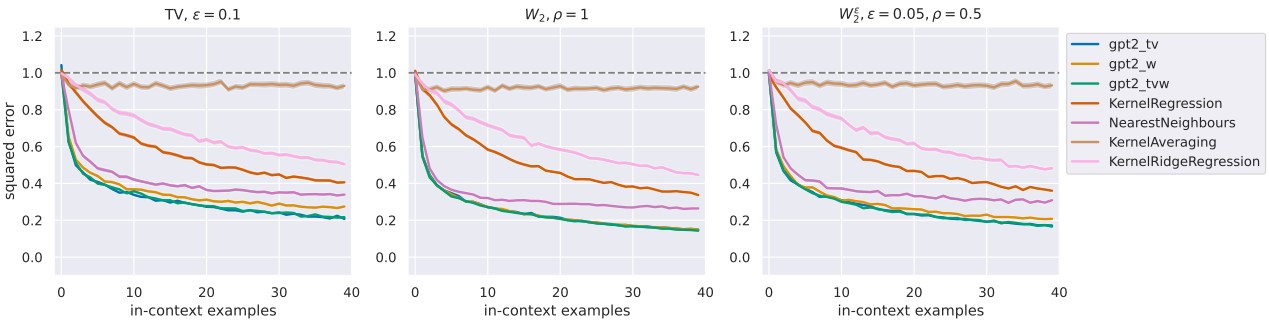

*Figure 3.* Results of the experiments on the class of functions based on the RBF kernel. The x-axis is the number of in-context samples, and the y-axis is the L2 loss of the predictor. The dashed grey line is the loss of the trivial predictor, $f = 0$. (Left) Result of the TV distance-based DRO. (Middle) Result of the Wasserstein distance-based DRO. (Right) Result of the outlier-robust Wasserstein distance-based DRO.

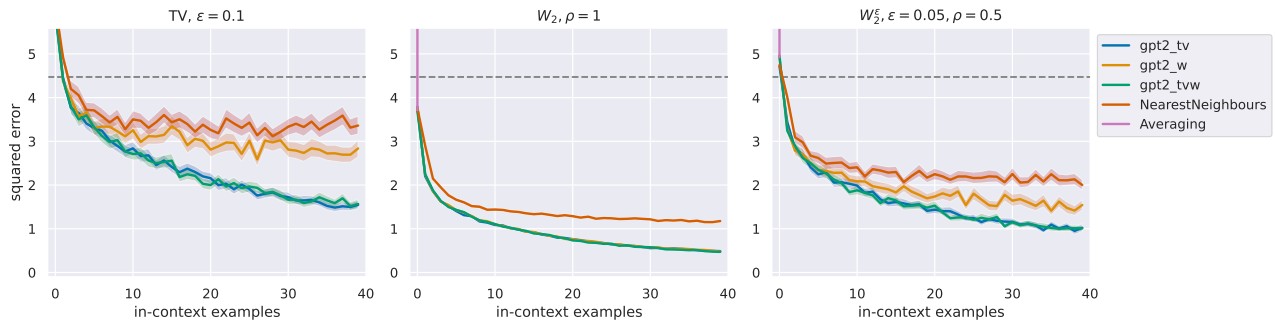

*Figure 4.* Results of the experiments on the class of shallow neural networks. The x-axis is the number of in-context samples, and the y-axis is the L2 loss of the predictor. The dashed grey line is the loss of the trivial predictor, $f = 0$. (Left) Result of the TV distance-based DRO. (Middle) Result of the Wasserstein distance-based DRO. (Right) Result of the outlier-robust Wasserstein distance-based DRO.

DRO problem and the ideal model has nearly the same performance. This indicates that the trained transformer works ideally; it approximately identifies the perturbed samples in a given in-context dataset, filters them out, and uses only the remaining samples for the mean and variance prediction. This indication is also consistent with the visualisation of the attention weight statistics of the trained transformer shown in Figure 6; the attention pattern for perturbed samples (columns with the red line) is significantly different from that for clean samples (columns without the red line), e.g., the first head (bottom) in Layer 2 of left figure and the last head (top) of Layer 2 in the right figure.

## 5. Related Works

**In-Context Learning** The base of our work aligns with Garg et al. (2022), which shows that the small transformer models (GPT2) can perform in-context learning when trained on tasks such as linear regression or decision forest prediction. Several researches followed this line of work by varying several aspects, like how the prediction changes as the diversity of pretraining diversity changes (Raventós et al., 2023). Raventós et al. (2023) found that in the sparse training sample regime, the transformer model predicts according to the dMMSE algorithm which is Bayes optimal on discrete distribution, whereas in the dense training sample regime, the transformer model predicts according to the Ridge regression model which is again Bayes optimal on continuous, Gaussian distribution. Our results in Section 4 extend this result to the DRO problem, the ideal model will perform the Bayes optimal prediction on the clean distribution according to Raventós et al. (2023)'s result, and since our transformer model trained on the DRO problem performs nearly the same as the ideal model, we can infer that the transformer model is predicting according to the Bayes optimal prediction under the distributionally-robust setting.

Theoretically, Von Oswald et al. (2023) has shown that shallow transformer models can perform the linear regression when trained by gradient flow, while not able to adapt to the covariate shift even trained on the mixture of the training distribution. This result aligns with Garg et al. (2022)'s observation, that the standard transformer model's performance drops significantly under the covariate shift. This problem is crucial in the application of distributionally-robust optimisation, where the training distribution itself should be a mixture of several distributions, otherwise, the model can memorise the single true distribution, not performing DRO. By introducing the dropout on the attention weights, we observe that such behaviour can be mitigated, and the transformer model can perform the DRO adapting to the mixture of the training distribution.

**Distributionally-Robust Optimisation**

DRO was suggested as a theoretically well-founded method to make the model robust to the distributional shift, especially in the adversarial setting. We discuss two main types of DRO solutions, the duality-based method and the resilience-based method. In Gao & Kleywegt (2016), which belongs to the first type, a general DRO problem was handled, which satisfies the so-called growth condition. One of the practical examples satisfying this condition is the L1 linear regression under Wasserstein distance-based DRO. From the duality, the worst-case distribution can be characterised by the dual problem, and the min-max solution can be rewritten as regularised empirical risk minimisation. While this method gives optimal min-max solution of Equation 1, their method is limited by the growth condition, making L2 regression or kernel regression out of scope. Also, due to its reliance on a convex programming solver, the method is not scalable to large-scale problems. An example of the second type of DRO solution is Zhu et al. (2022), where the idea of resilience is introduced to control the effect of the perturbation on the loss function. The resilience property is defined as the Lipschitz-ness property of the loss of ERM solution w.r.t. the perturbation of the distribution. By introducing the minimum distance distribution, which is the distribution that is in a pre-specified family of distribution, and has the minimum distance to the perturbed distribution, one can show that the loss on the true distribution is close to the loss on the minimum distance distribution, which is again close to the loss on the perturbed distribution, by the resilience property. In general such minimum distance distribution can not be computed in closed form, hence the method usually relies on the iterative refinement to weigh the training data and certify the resilience property. Both approaches are limited in practice, the duality-based method is hard to extend to the general models and while the resilience-based method is more general, its computational complexity is scaled as $d^8$ where $d$ is the dimension of the input space.

Unlike the existing works, our work is based on empirically training the transformer model on the DRO problems, which can be seen as the extension of the adversarial training framework. Sinha et al. (2018) introduced the general procedure for the non-linear model to be robust to the adversarial perturbation, via adversarial training. The adversarial training procedure depends on the types of perturbations they allow, which is based on the dual formulation of Equation 1, similar to Gao & Kleywegt (2016)'s work. Unlike most of the existing works, Sinha et al. (2018) provides the theoretical guarantee for the robustness of the model using the statistical learning theory.

## 6. Conclusion and Future Works

In this paper, we proposed the use of in-context learning for the distributionally robust optimisation, showing that they

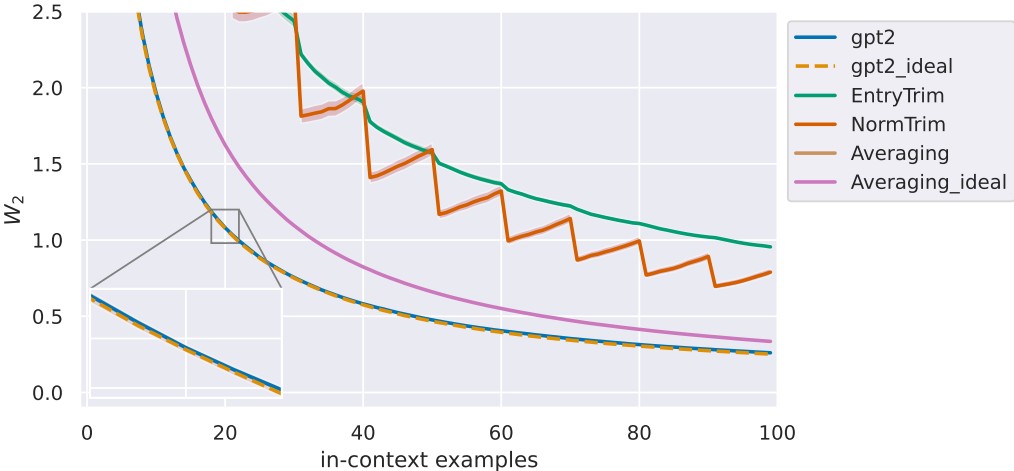

*Figure 5.* Results of the experiments on the mean and variance estimation. The x-axis is the number of in-context samples, and the y-axis is the Wasserstein-2 distance between the predicted and the true mean and variance.

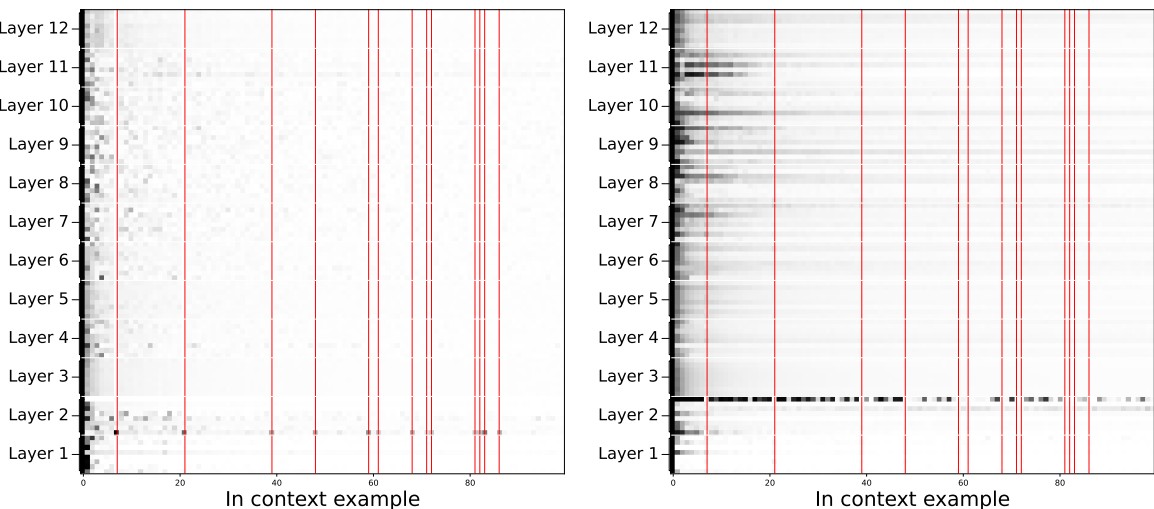

*Figure 6.* (Left) An attention weight from $i$-th in-context sample to itself. (Right) An attention value from $i$-th in-context sample to the first sample. The red line indicates the perturbed samples. The x-axis is the index of in-context samples, and $y$-axis is the index of layer and heads.

can be applied to various types of perturbations and models, extending the existing works on DRO. Specifically, we show three basic classes of perturbations, each represented by TV distance, the Wasserstein distance, and the outlier-robust Wasserstein distance can be handled by the transformer model. We also show that the transformer can be used for the non-linear class of functions, which is more flexible than the existing works on DRO. We also provide the preliminary interpretation of the success of the transformer in DRO, by showing that the transformer's performance on the mean and variance prediction performance matches with the ideal model that filters out the perturbed samples.

One of the future directions is extending our work to more complex problems. Some of the recent works on DRO consider more practical problems like decision problems including contextual bandit (Si et al., 2020) or imitation learning (Bashiri et al., 2021), which suggests that the use of transformer for RL problems like decision transformer (Chen et al., 2021) can be a good direction for the future work. The other future works include extending the types of perturbation that we allow, including Sinkhorn distance (Wang et al., 2022) and the Kernel MMD (Staib & Jegelka, 2019; Zhu et al., 2021b).

Another future direction is to give theoretical guarantees for the transformer's performance in DRO. While we empirically show that the transformer can perform the DRO, the theoretical guarantee is still missing. One possible direction is using the adversarial training framework, and deriving the theoretical guarantee for DRO in a similar manner of Sinha et al. (2018). Also, recent theoretical works on in-context learning (Von Oswald et al., 2023; Ahn et al., 2023) suggest that transformers implement the gradient-based mesa-optimiser. Extending this line of work to our setting can suggest what algorithm the transformer is implementing, which gives interpretable results that can be used to analyse the transformer's performance in DRO.

## Acknowledgements

This work was supported by the National Research Foundation of Korea (NRF) grant funded by the Korean Government (MSIT) (No. RS-2023-00279680).

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

*Table 1.* Hyperparameters for the experiments.

| Hyperparameter | Value |
|---|---|
| Batch size | 64 |
| Learning rate | 1e-4 |
| Training step | 500K |
| Optimiser | Adam |
| Attention dropout rate | 0.1 |
| Residual / Embedding dropout rate | 0.0 |
| Number of layers | 12 |
| Number of heads | 8 |
| Embedding dimension | 256 |

# A. Experiment Details

All the training and evaluations were done on four NVIDIA RTX A5000 GPUs, with two Intel Xeon Gold 6234 CPUs with 256GB of RAM. For TV distance-based DRO, the training takes around 10 GPU hours for a single model, whereas the Wasserstein distance-based DRO takes around 12 GPU hours, due to the complexity of the Wasserstein distance computation. The evaluations were done on the same machine and takes less than 1 GPU hour for all the experiments.

All our experiments assume that the input dimension is 20 and the output dimension is 1. We limit the number of points in a single sequence to 40 for regression task and 100 for mean and variance estimation task. As in Garg et al. (2022), we employ the curriculum learning schedules, where we begin from the input dimension 10, number of points 20, and the distribution close to isotropic at the beginning, and gradually increase the input dimension, number of points, and the anisotropy of the distribution. In detail, the training data generation is described in Algorithm 1.

---
**Algorithm 1** Data point generation for the regressions.

---
1: **Input :** $d$ the input dimension, $N$ the number of points, $\alpha$ the anisotropy hyperparameter.
2: Sample the anisotropy parameter $r \sim \beta(\alpha, 1)$.
3: Sample the mean $\mu \sim \mathcal{N}(0, I_d)$.
4: Sample the isotropic noise $z^{(i)} \sim \mathcal{N}(0, I_d)$.
5: Sample the anisotropic covariance $\Sigma \sim \text{Wishart}(I_d/d, d)$.
6: Sample the anisotropic noise $w^{(i)} \sim \mathcal{N}(0, \Sigma)$.
7: Create data by $x^{(i)} = \mu + rz^{(i)} + (1 - r)w^{(i)}$.

---

The training data generation of the experiment in Figure 5 is slightly different from the other experiments. The training data generation is described in Algorithm 2.

---
**Algorithm 2** Data point generation for the mean and variance estimation.

---
1: **Input :** $d$ the input dimension, $N$ the number of points, $\alpha$ the anisotropy hyperparameter.
2: Sample the anisotropy parameter $r \sim \beta(\alpha, 1)$.
3: Sample the anisotropic standard deviation $\overline{\sigma}_j \sim \Gamma(2, 2)$ for $j = 1, \ldots, d$.
4: Compute the standard deviation as $\sigma_j = (1 - r) \cdot \overline{\sigma}_j + r$.
5: Sample the mean $\mu \sim \mathcal{N}(0, I_d)$.
6: Sample the data $x^{(i)} \sim \mathcal{N}(\mu, \text{diag}(\sigma))$.

---

The details on the hyperparameters are given in Table 1.

# B. Additional Experiments

In this section, we report the results of the experiments on the out-of-distribution cases, where the strength of the perturbation is weaker or stronger than the training strength. In Figure 7 and 8, we report the risk of the models trained on the linear functions with the different strengths of the perturbation. The gpt2_id models are the models trained on in-distribution, i.e.,

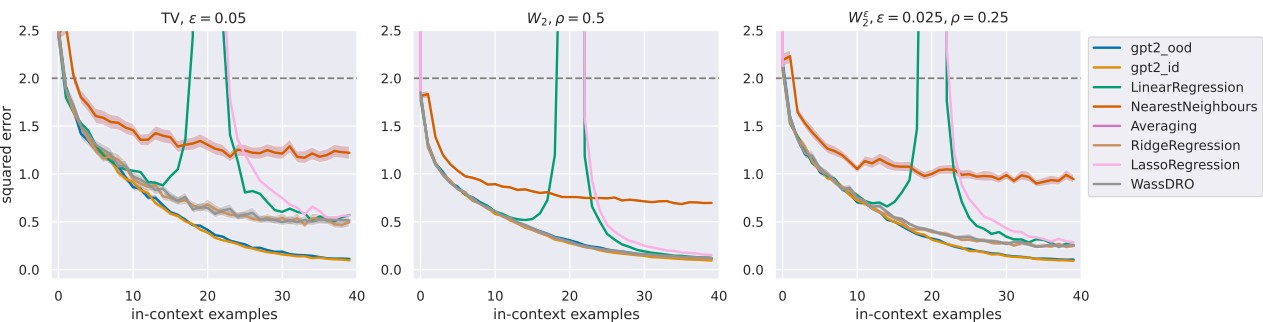

*Figure 7.* Results of the experiments on the class of linear functions with the weaker perturbation. (Left) Result of the TV distance-based DRO. (Middle) Result of the Wasserstein distance-based DRO. (Right) Result of the outlier-robust Wasserstein distance-based DRO.

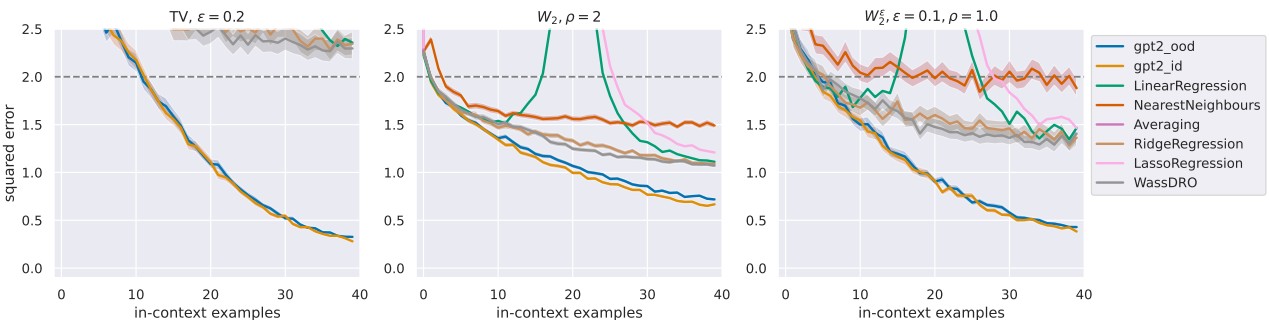

*Figure 8.* Results of the experiments on the class of linear functions with the stronger perturbation. (Left) Result of the TV distance-based DRO. (Middle) Result of the Wasserstein distance-based DRO. (Right) Result of the outlier-robust Wasserstein distance-based DRO.

in Figure 7, they are trained on the TV distance-based DRO with the strength $\epsilon = 0.05$, Wasserstein distance-based DRO with the strength $\rho = 0.5$, and the outlier-robust Wasserstein distance-based DRO with the strength $\epsilon = 0.025$ and $\rho = 0.25$, respectively, and in Figure 8, they are trained on the TV distance-based DRO with the strength $\epsilon = 0.2$, Wasserstein distance-based DRO with the strength $\rho = 2.0$, and the outlier-robust Wasserstein distance-based DRO with the strength $\epsilon = 0.1$ and $\rho = 1.0$, respectively. The gpt2_ood models are the models trained on out-of-distribution, in both Figures, they are trained on the TV distance-based DRO with the strength $\epsilon = 0.1$, Wasserstein distance-based DRO with the strength $\rho = 1.0$, and the outlier-robust Wasserstein distance-based DRO with the strength $\epsilon = 0.05$ and $\rho = 0.5$, respectively. Surprisingly, the gpt2_ood model, which is the model trained on the different strengths of perturbation, performs the same in both weaker and stronger perturbation cases, indicating that the transformer model can generalise to the out-of-distribution cases.

## C. Brief Introduction to DRO

In this section, we give a brief introduction to the distributionally-robust optimisation and prove that our Algorithm 3 is a valid perturbation for the Wasserstein distance-based DRO.

Distributionally-robust optimisation (DRO) is a method to make the model robust to the distributional shift, especially in the adversarial setting. The DRO problem considers the combination of adversarial optimisation and stochastic optimisation, defined as

$$\text{(DRO)} \quad \min_{\theta \in \Theta} \max_{p \in \mathcal{P}} \mathbb{E}_{z \sim p} \mathcal{L}(\theta; z),$$

where $\mathcal{L}(\theta; z)$ is the loss function, $\Theta$ is the parameter space. Here the $\mathcal{P}$ is usually called the ambiguity set, which defines

the distributions that the model should generalise to. There is no restriction on the form of the ambiguity set, where we focus on the balls in the probability spaces, i.e.,

$$\mathcal{P} = \{q \in \mathcal{P}(\mathbb{R}^d) : D(p, q) \leq \epsilon\}$$

for the metric $D$ on the probability space. The ambiguity set of these forms determines the types of perturbations that the result of the DRO can be robust to, and the choice of the ambiguity set is crucial in the DRO. Two of the most well-known ambiguity sets are based on the TV distance and the Wasserstein distance.

Total variation distance is defined as

$$\mathrm{TV}(p, q) = \sup_{A \in \mathcal{A}} |p(A) - q(A)|$$

where $\mathcal{A}$ is the set of all the measurable sets. If $p, q$ has the densities, then the TV distance can be written as

$$\mathrm{TV}(p, q) = \frac{1}{2} \int |p(x) - q(x)| dx$$

and it also allows some dual formulation, i.e.,

$$\mathrm{TV}(p, q) = \sup_{f : \|f\|_\infty \leq 1} \int f(x)(p(x) - q(x)) dx.$$

The TV distance can be understood as moving the densities without any restrictions, up to the proportion $\epsilon$. For example, if $p$ and $q$ has disjoint supports, then

$$\mathrm{TV}(p, (1 - \epsilon)p + \epsilon q) = \epsilon.$$

The TV distance-based DRO can be understood as outlier-robust optimisation, where the model should be able to handle outliers that can be arbitrarily far from the training distribution, up to some proportion of the training distribution.

The $p$-Wasserstein distance is defined as

$$W_p(\mu, \nu) = \inf_{\pi \in \mathcal{P}(\mu \times \nu)} \left( \int d(x, y)^p d\pi(x, y) \right)^{1/p}$$

where $d(x, y)$ is the metric on the space, and $\mathcal{P}(\mu \times \nu)$ is the set of all the couplings of two measures $\mu, \nu$, i.e., $p \in \mathcal{P}(\mathbb{R}^d \times \mathbb{R}^d)$ is the coupling of $\mu, \nu$ if

$$p(A \times \mathbb{R}^d) = \mu(A), \quad p(\mathbb{R}^d \times B) = \nu(B).$$

The Wasserstein distance can be understood as the minimum cost of transporting the mass from $\mu$ to $\nu$, where the cost depends on the metric $d$. The Wasserstein distance-based DRO can be understood as the geometrically robust optimisation, where the model should be able to handle the distributional shift that can be transported to the training distribution with the cost of $\epsilon$.

The special case $p = 2$ which we focus on in this paper, has several nice properties that we can leverage.

**Theorem C.1** (Breiner's theorem). *Let $X, Y$ be the closure of bounded open sets in $\mathbb{R}^d$ and $\mu, \nu$ be the probability measures on $X, Y$ respectively. If $\mu$ is absolutely continuous w.r.t. the Lebesgue measure, then the optimal coupling $\pi^*$ between $\mu$ and $\nu$ can be written as*

$$\pi^* = (Id \otimes T)_{\#}\mu$$

*where $T$ is a measurable function such that $T_{\#}\mu = \nu$ and $T(x) = \nabla u(x)$ for some convex function $u$ and we say this map is the optimal transport map.*

**Lemma C.2** (Breiner's theorem for discrete distribution). *Suppose that $\mu, \nu$ are discrete random variables with $N$ particles each, i.e.,*

$$\mu = \frac{1}{N} \sum_{i=1}^{N} \delta_{x_i}, \quad \nu = \frac{1}{N} \sum_{i=1}^{N} \delta_{y_i},$$

*then the optimal coupling $\pi^*$ between $\mu$ and $\nu$ can be written as*

$$\pi^* = (I \otimes T)_{\#}(\mu)$$

*where $T$ is a bijection between the set of particles, i.e., $T(x_i) = y_j$ for some $j$.*

Breiner's theorems allow us to understand the $W_2$ optimal transport plan as a bijection between the particles, which allows us to construct the perturbation by transporting the points. However, the generated perturbation can be too strong to be contained in the Wasserstein ball, which requires rejection sampling to be contained in the Wasserstein ball. Instead, we use displacement interpolation, which is the interpolation between the optimal transport map and the identity map, so that one can scale the transport plan to be contained in the Wasserstein ball. This is known as McCann's displacement interpolation, which is given in the following theorem.

**Theorem C.3** (McCann's displacement interpolation). *Let $\mu, \nu$ be the probability measures on $\mathbb{R}^d$ and $T$ be the optimal transport map between them with $\pi^*$ the optimal transport plan. Define $\pi_t : \mathbb{R}^d \times \mathbb{R}^d \to \mathbb{R}^d$ for $0 \leq t \leq 1$ as*

$$\pi_t(x, y) = (1 - t)x + ty,$$

*then $(\pi_t)_{\#}\pi^*$ is displacement interpolation between $\mu$ and $\nu$, i.e.,*

$$W_2(\mu, (\pi_t)_{\#}\pi^*) = tW_2(\mu, \nu).$$

### C.1. Creating Perturbation

**TV distance** The TV distance is the simplest perturbation to implement. From its definition

$$\mathrm{TV}(p, q) = \int |p(x) - q(x)| dx,$$

if two distribution $p_1, p_2$ has disjoint support, then

$$\mathrm{TV}(p_1, (1 - \epsilon)p_1 + \epsilon p_2) = \epsilon$$

and if they do not have disjoint support, then

$$\mathrm{TV}(p_1, (1 - \epsilon)p_1 + \epsilon p_2) \leq \epsilon.$$

Leveraging this inequality, we sample another distribution $q$ from the family of distribution $\mathcal{Q}$, and replace the $(x^{(i)}, y^{(i)})$ with probability $\epsilon$.

**W2-distance** We recall the definition of the Wasserstein distance first.

**Definition C.4** (Wasserstein distance). The Wasserstein distance on the metric space $(X, d)$ is the distance between two distribution $\mu, \nu$ defined as

$$W_{p,q}(\mu, \nu) = \inf_{\pi \in \mathcal{P}(\mu \times \nu)} \left( \int d(x, p)^p d\pi(x, y) \right)^{q/p}$$

where the infimum is taken over all the couplings of two measures $\mu, \nu$, which are the measures on the product space $X \times X$ whose marginals are $\mu$ and $\nu$, i.e.,

$$\pi(A \times X) = \mu(A), \quad \pi(X \times B) = \nu(B).$$

In our paper, we focus on the case when $p = 2, q = 2$, which is the well-known 2-Wasserstein distance. Note that extending it to arbitrary $q$ is straightforward.

This Wasserstein distance satisfies the following property, which we leverage to create the $\epsilon$-perturbation.

**Lemma C.5.** *Let $p$ and $q$ be two distributions on $\mathbb{R}^d$ that are either both continuous, or discrete and satisfies*

$$p = \frac{1}{N} \sum_{i=1}^{N} \delta_{x_i}, \quad q = \frac{1}{N} \sum_{i=1}^{N} \delta_{y_i},$$

*then the optimal coupling between $p$ and $q$ can be written as*

$$\pi^* = (I \otimes g_{\#})(p).$$

*Moreover, the distribution*

$$q_\delta = ((1 - \delta)\mathrm{id} + \delta(g - \mathrm{id}))_{\#}p$$

*satisfies*

$$W_{2,2}(p, q_\delta) = \delta W_{2,2}(p, q).$$

---

**Algorithm 3** Perturbation for $W_{2,2}$ distance.

---

1: **Input :** $\{x^{(i)}\}_{i=1}^N$ the training samples, $\epsilon$ the perturbation parameter.
2: Sample $\{z^{(i)}\}_{i=1}^N$ from other distribution $q$.
3: Compute $d = W_{2,2}(\frac{1}{N}\sum_{i=1}^N \delta_{x^{(i)}}, \frac{1}{N}\sum_{i=1}^N \delta_{z^{(i)}})$ and the optimal pushforward map $g$ between them via Hungarian algorithm.
4: Define $q_\delta$ as in Lemma C.5, with $\delta = \sqrt{\epsilon/d}$.
5: **Return** $\{q_\delta(x^{(i)})\}_{i=1}^N$.

---

Our procedure for creating the perturbation is given in Algorithm 3.

**Outlier-robust Wasserstein distance**

The outlier-robust Wasserstein distance is defined as

$$W_p^\epsilon(\mu, \nu) = \inf_{\substack{\mu' \in \mathcal{P}(\mathbb{R}^d) \\ \mathrm{TV}(\mu,\mu') \leq \epsilon}} W_p(\mu', \nu).$$

One can think of this as the sequential application of the TV distance-based perturbation, then the Wasserstein distance-based perturbation, thus allowing both types of perturbations.

We adopt the Setting B of Nietert et al. (2023), where one can understand the outlier-robust Wasserstein distance $W_2^\epsilon$ based DRO as the sequential application of the $W_2$ then the TV distance-based DRO. Therefore, we apply Algorithm 3 first, then apply the procedure described in TV distance-based DRO.

