# OpenReview forum: "Transformers Can Perform Distributionally-robust Optimisation through In-context Learning"
_ICML.cc/2024/Workshop/ICL — ICML 2024 Workshop ICL Poster_

### Official Review · Reviewer_6EBC · 2024-06-07
**An interesting empirical look at in-context learning for distributionally-robust optimization**

**Rating:** 3
**Fit:** 3
**Confidence:** 2

**Workshop Review:**

This paper presents a study of a Transformer model capability of performing distributionally-robust optimization via in-context learning. It analyzes several underlying function families ranging from very broadly studied linear models to functions based on RBF kernels and simple shallow neural networks. While the capability of the Transformer model to solve noisy regressions problems via in-context learning has already been studied in prior literature and is not entirely novel, this paper discusses a more general distributionally-robust framework. Furthermore, it makes an interesting observation that perturbations with small total-variation distance can be very detrimental to performance of the simplest classical regression approaches, but a more complex Transformer model can learn to detect and discard outliers. These observations are intuitive, but appear to be sufficiently novel and may be of potential interest to a broader community.

The study is leaning towards empirical analysis, but while the exact mechanisms behind the performance of a learned Transformer are not investigated, experimental results seem to support a hypothesis that the Transformer learns to ignore outliers in the TV case.

The paper is well written and appears to be correct. It motivates and discusses the problem very clearly. There are a few minor misprints, but no major issues.

**Reason For Not Giving Higher Score:**

Originally, I was not entirely satisfied with the discussion of generalization capabilities of learned in-context algorithms, but I later discovered a more in-depth analysis in Appendix B. I think the paper would benefit from adding a very brief summary of the Appendix B setup and/or observations in the main text.

As a minor comment, it would be helpful if the empirical results shown in Figures 2-4 were discussed in a bit more detail (maybe in Appendix). Figure 5 is difficult to interpret because: (a) it is difficult to see the difference between two curves, we expect that the Transformer should perform at least a little worse than the oracle-based method and it would be useful to see this in the plot (perhaps the authors could provide another plot, or an inset showing the difference between the two?); (b) it would also be useful to show another method to see the similarity between the curves more clearly (the third curve will be very distinct and this will highlight the fact that the Transformer curve is almost indistinguishable from the oracle).

**Reason For Not Giving Lower Score:**

Overall, a detailed and well-executed empirical study of distributionally-robust in-context learning in Transformers. I did not have any major issues with the publication. The paper considers multiple models and types of perturbations. I also appreciated the discussion of model generalization and Section 4 presenting an insightful look at a potential mechanism behind the learned DRO algorithm.

---

### Official Review · Reviewer_iwuA · 2024-06-11
**Good paper, sound contribution, along with some interesting insights**

**Rating:** 2
**Fit:** 3
**Confidence:** 2

**Workshop Review:**

The paper is well-written and well-motivated, and it explores a promising and novel research direction by investigating the application of in-context learning to distributionally robust optimization. Building on the results from Garg et al. (2022), the paper extends the analysis to cases where each example (both input and output) is perturbed. The empirical results (linear regression, RBF kernel, and shallow neural network) demonstrate that in-context learning through Transformers can effectively perform distributionally robust optimization. Additionally, the paper provides a very insightful analysis, showing that the trained model can differentiate between “clean” and corrupted samples. With the above comments, I recommend a clear acceptance of the paper.

**Reason For Not Giving Higher Score:**

The choice of noise thresholds (0.1, 1.0, 0.5) seems arbitrary to me. The results would be easier to interpret if these thresholds were determined based on the true noise of the data. In other words, it would be interesting to see how robust transformers are to noise by determining the threshold at which the results no longer hold.

**Reason For Not Giving Lower Score:**

The contribution of the paper is sound and directly aligns with the interests of the workshop.

---

### Author Response · Authors · 2024-06-20
**Update on camera ready version and answers to several questions**

We thank the reviewers and area chair for the constructive comments.

We updated our camera-ready version based on the comments from the reviewers.
Here is the summary of the updates.
- We fixed several typos in the paper.
- In the main text of the paper, we added the summary of the results in Appendix B as suggested by the review 6EBC so that the reader can understand our out-of-distribution results better.
- We modified Figure 5 to include four naive baselines (a non-robust one, two robust baselines, and an ideal baseline), and also an inset zooming in the range with 18-22 in context samples, which shows that gpt2 with an oracle performs slightly better than gpt2 without an oracle.

Regarding the comment of reviewer iwuA, it is true that our choice of noise thresholds is arbitrary, and does not arise from some practical application. But we want to point out that Appendix B contains results with two different choices of thresholds. Specifically, it contains the results for weaker perturbations (half threshold) and stronger perturbations (twice threshold). Although not perfect, these three options suggest that our method works for a range of perturbation strengths.

---

### Meta-Review · Area_Chair_PyXD · 2024-06-12

**Recommendation:** 2

**Metareview:**

This paper explores the application of transformers in performing distributionally robust optimization (DRO) through in-context learning. The authors present empirical evidence demonstrating that transformers can handle DRO tasks, even with multiple types of perturbations.

Both reviewers agree that the paper is well-written and fits well within the workshop's scope. The empirical results are solid and extend the understanding of transformers' capabilities in DRO contexts. Reviewer iwuA’s concern about the arbitrary choice of noise thresholds and Reviewer 6EBC’s suggestion to improve the clarity of the figures and appendices are valuable feedback.

Based on the reviews and additional insights, I recommend the paper for acceptance as a poster presentation.

---

### Decision · Program_Chairs · 2024-06-17

Accept (Poster)